# Mitigating Doxorubicin-Induced Cardiotoxicity through Quercetin Intervention: An Experimental Study in Rats

**DOI:** 10.3390/antiox13091068

**Published:** 2024-08-31

**Authors:** Patricia Lorena Dulf, Camelia Alexandra Coadă, Adrian Florea, Remus Moldovan, Ioana Baldea, Daniel Vasile Dulf, Dan Blendea, Adriana Gabriela Filip

**Affiliations:** 1Faculty of Medicine, Iuliu Haţieganu University of Medicine and Pharmacy, 400012 Cluj-Napoca, Romania; cimpan.patricia@umfcluj.ro (P.L.D.); dulf.daniel@umfcluj.ro (D.V.D.); 2Department of Molecular Sciences, Iuliu Haţieganu University of Medicine and Pharmacy, 400012 Cluj-Napoca, Romania; aflorea@umfcluj.ro; 3Department of Functional Biosciences, Iuliu Haţieganu University of Medicine and Pharmacy, 400012 Cluj-Napoca, Romania; moldovan.remus@umfcluj.ro (R.M.); gabriela.filip@umfcluj.ro (A.G.F.); 4Municipal Clinical Hospital, 400139 Cluj-Napoca, Romania; 5Internal Medicine Department, Faculty of Medicine, Iuliu Haţieganu University of Medicine and Pharmacy, 400012 Cluj-Napoca, Romania; dblendea@me.com; 6Department of Cardiology, Heart Institute, 400001 Cluj-Napoca, Romania

**Keywords:** antioxidants, cardiotoxicity, quercetin, reactive oxygen species, microscopy, doxorubicin, oxidative stress, antineoplastic agents, DNA damage

## Abstract

Doxorubicin (DOX) is an effective anticancer drug, but its use is limited by dose-dependent heart toxicity. Quercetin is a natural antioxidant frequently studied for its beneficial properties. Moreover, a wide range of dietary supplements are available for human use. This in vivo study aimed to explore the potential cardioprotective effects of quercetin in chronic DOX treatment. A total of 32 Wistar rats were randomly divided into four groups: control, DOX, DOX/Q-50, and DOX/Q-100, treated with saline, 2.5 mg/kg body-weight DOX, 2.5 mg/kg body-weight DOX + 50 mg quercetin, and 2.5 mg/kg body-weight DOX + 100 mg quercetin, respectively, for two weeks. Rats were monitored using cardiac ultrasound (US) and markers for cardiac injury. Oxidative damage and ultrastructural changes in the heart were investigated. Chronic DOX treatment led to a decline in cardiac function and elevated values of NT pro-BNP, troponin I, and CK-MB. Quercetin treatment slightly improved certain US parameters, and normalized serum NT pro-BNP levels. Furthermore, DOX-induced SOD1 depletion with consequent Nrf2 activation and DNA damage as shown by an increase in γH2AX and 8HOdG. Quercetin treatment alleviated these alterations. Oral administration of quercetin alleviated serum markers associated with DOX-induced cardiotoxicity. Furthermore, it exhibited a favorable impact on the cardiac US parameters. This suggests that quercetin may have potential cardioprotective properties.

## 1. Introduction

Cardiovascular events linked to cancer treatments contribute to excess cardiovascular and oncological mortality, thereby significantly influencing prognosis and quality of life [1]. Even though most oncological treatment modalities may have some degree of cardiotoxic effects, chemotherapy-induced cardiotoxicity stands out as the primary cause of morbidity and mortality among cancer survivors [2,3].

Anthracyclines are potent antitumor agents used in a wide variety of solid organ tumors and hematologic malignancies, including leukemia, lymphoma, breast cancer, lung cancer, multiple myeloma, and sarcoma hematological cancers [4]. Although doxorubicin (DOX) has become one of the most effective chemotherapeutic agents, it was noted early on that its use was complicated by the development of heart failure [5].

Anthracycline-induced cardiotoxicity is categorized into two distinct types: Type I and Type II. Type I cardiotoxicity is characterized by permanent and irreversible structural myocardial damage. Type II on the other hand is reversible and allows for treatment discontinuation and restart after the recovery of the patients. It involves ventricular dysfunction without observable myocardial rearrangements [6,7,8]. The timeline for anthracycline-induced cardiac side effects varies significantly. Acute coronary syndromes, myopericarditis, and various ventricular arrhythmias can manifest within weeks of initiating treatment. Long-term survivors, initially asymptomatic during remission, may later exhibit signs of systolic and diastolic dysfunction, raising concerns about potential progression to severe congestive heart failure [6]. The risk of cardiac side effects, including death, sharply increases when the cumulative dose of DOX reaches 550 mg/m^2^. Moreover, asymptomatic cardiac dysfunction can persist in up to 57% of cancer patients in Western countries years after anthracycline chemotherapy [9].

Significant research efforts have been made to explore treatment modalities aimed at preventing and potentially reversing cardiovascular damage caused by chemotherapeutic drugs [10]. Despite this, the only cardio-protective agent currently approved for clinical use is dexrazoxane. Dexrazoxane operates by chelating iron, a key player in anthracycline-induced oxidative stress, thereby preventing the formation of free radicals and subsequent damage to cardiac tissues. It is important to note, however, that dexrazoxane treatment does not entirely eradicate the risk of cardiac toxicity induced by anthracyclines [11]. Thus, ongoing research endeavors seek to unravel additional cardio-protective modalities that could complement or surpass the efficacy of dexrazoxane.

In this context, antioxidants, dietary polyphenols, and other natural antioxidants have rapidly gained attention as viable candidates for cardio-protection [12]. Quercetin, a naturally occurring flavonoid abundant in various fruits, vegetables, and plant-based foods, has garnered attention for its potential cardioprotective effects [13]. Studies suggest that quercetin possesses antioxidant and anti-inflammatory properties, which contribute to its ability to mitigate cardiovascular damage. Moreover, quercetin may play a crucial role in protecting the heart by reducing oxidative stress, inflammation, and apoptosis, thus safeguarding against cardiotoxicity induced by various stressors, including chemotherapeutic agents [14,15,16]. Additionally, quercetin has been implicated in improving endothelial function, modulating lipid metabolism, and enhancing the overall cardiovascular system’s resilience [14]. While further investigation is needed to elucidate the precise mechanisms and optimal dosage for cardioprotection, the accumulating evidence underscores quercetin’s potential as a promising natural compound in promoting heart health and preventing cardiovascular disorders [17,18].

Thus, the current study aimed to investigate the potential protective effects of different quercetin doses on chronic cardiac toxicity induced by DOX. Additionally, we aimed to assess whether quercetin treatment can improve heart output and restore DOX-induced damage at molecular and ultrastructural levels. Secondly, we sought to explore the impact of quercetin on the main mechanisms of chronic DOX-induced cardiotoxicity and cardiac ultrastructure.

## 2. Materials and Methods

### 2.1. Study Design

This analytical, experimental study was performed on 32 Wistar rats weighing 150–250 g, which were housed in the animal facility of “Iuliu Hațieganu” University of Medicine and Pharmacy, Cluj-Napoca, Romania with stable temperatures of 23 ± 2 °C and a 12 h dark/light cycle, and received food and water ad libitum. Rats were acclimatized to laboratory conditions for a period of one week prior to the study initiation. The number of animals was chosen based on the principles of the 3R’s for animal studies [19].

Animals were divided randomly into 4 study groups, of 8 rats each, as follows: (1) the control group, receiving six intraperitoneal (i.p.) injections of 0.9% saline solution; (2) the DOX group, receiving six i.p. injections with 2.5 mg/kg body-weight doxorubicin; (3) the DOX/Q-50 group, receiving six i.p. injections with 2.5 mg/kg body-weight doxorubicin and 50 mk/kg/day quercetin orally; and (4) the DOX/Q-100 group, receiving six i.p. injections with 2.5 mg/kg body-weight doxorubicin and 100 mk/kg/day quercetin orally (Figure 1). DOX injections were administered at a distance of 48 h between them for a total treatment time of two weeks, to a cumulative dose of 15 mg/kg body-weight [9], while the control arm received the same number of injections but with saline solution. Quercetin was administered daily by oral gavage at concentrations of 50 mg/kg/day and 100 mg/kg/day in the DOX/Q-50 and DOX/Q-100 arms, respectively. After 48 h from the last treatment administration, the animals were subjected to ultrasound examination and then euthanized under an i.p. ketamine and xylazine (90 mg/kg body-weight; 10 mg/kg body-weight) anesthesia. Blood and heart tissues were collected from each rat for further processing.

The selection of the animal population was determined to balance the need for a sufficient number of samples to attain statistical significance while keeping it ethically minimal. The calculation of the sample size was conducted based on prior pilot studies, setting a beta value of 0.8 and an alpha value of 0.05.

### 2.2. Rat Cardiac Echocardiography

Cardiac ultrasound (US) investigations were conducted at the conclusion of the two-week experiment using an Ultrasonix ultrasound machine equipped with a 15–20 Hz phased array probe (12S-RS), specifically designed for rat studies. Prior to the procedure, the animals were anesthetized with the same doses of ketamine and xylazine. The anterior chest hair of all animals was shaved, and they were positioned in the left lateral decubitus on a heated plate set at 37 °C to maintain the rats’ body temperature. To prevent the formation of bubbles and potential examination artifacts, prewarmed ultrasonography gel was generously applied to both the probe and the rats’ chests. The examination settings included a transmission frequency of 15 MHz, a depth of 1.5 cm, and a frame rate of 25 frames per second. M-mode echocardiography was employed for 2D image acquisition and measurements from the parasternal long-axis view at the mid-papillary muscle level. The measured parameters encompassed systolic and diastolic left ventricular dimensions (LVDs, LVDD). Fractional shortening (FS) was computed from the internal diameters of the left ventricle during systole and diastole. The left ventricular ejection fraction (LVEF) was determined using the Teicholz method. To ensure accuracy, a single investigator conducted a minimum of four measurements for each rat, averaging the results to obtain the final recorded value. All measurements were performed following image stabilization and adhered to the leading-edge method of the American Society of Echocardiography [20].

### 2.3. Biochemical Assays

To evaluate the cardiotoxicity of DOX, N-terminal pro-brain natriuretic peptide (NT-proBNP) (Elabscience, Houston, TX, USA, cat: E-EL-R3023) from serum and creatine kinase-MB (CK-MB) (Elabscience, Houston, TX, USA, cat: E-CL-R0722) as well as Troponin I (TnI) (Elabscience, Houston, TX, USA, cat: E-EL-R1253) levels from heart tissues were quantified by ELISA according to the manufacturer’s protocol. Additionally, 8-HO-guanosine (8HOdG), a marker of oxidative DNA damage, was assessed from heart homogenates. The results were expressed as pg/mg protein.

### 2.4. Western Blot Analysis

Total protein lysates (40 µg/lane) from cardiac tissues were used to quantify SOD1, γH2AX, and Nrf2 by Western blot using the Bio-Rad Mini-protean system. Blots were blocked with StartingBlock Buffer (Thermo Fischer Scientific, Waltham, MA USA) and then incubated with primary antibodies from Santa Cruz Biotechnology INC (Dallas, TX, USA), targeting SOD-1 (G-11: sc-17767, 1:500 dilution) and Nrf2 (437C2a: sc-81342, 1:500 dilution). Double-strand DNA damage was detected using the γH2AX antibody (R20244, NSJ Bioreagents, San Diego, CA, USA, 1:1000 dilution). Membranes were washed and incubated with the corresponding secondary antibody linked with HRP, anti-mouse, 1:2000 (W4021, Promega, Madison, WI, USA), antirabbit, 1:1000, (#7074, CellSignaling Technology, Danvers, MA, USA) for 90 min, at room temperature. After washing, membranes were incubated with Supersignal West Femto-Chemiluminescent substrate (Thermo Fisher Scientific, Rockford, IL, USA) for the detection step. Image acquisition was done on ChemiDoc (Bio-Rad, Hercules, CA, USA), with quantification on Image Lab (Bio-Rad, Hercules, CA, USA). Beta-actin served as the protein loading control and for normalization.

### 2.5. Transmission Electron Microscopy

Transmission electron microscopy (TEM) analysis was conducted on samples of left ventricular myocardium. Samples were prefixed for 2 h with 2.7% glutaraldehyde (Agar Scientific, Stansted, UK) and postfixed for 1.5 h with 1% OsO4 (Electron Microscopy Sciences, Hatfield, PA, USA) in 0.15 M phosphate buffer (pH = 7.4). After graded dehydration of samples with acetone (Merck, Darmstadt, Germany) (30–100%, 15–30 min each bath), they were infiltrated with a series of EMBED 812 (Electron Microscopy Sciences, Hatfield, PA, USA) epoxy resin in acetone (30–90% 1–2 h each bath, and 100% overnight). Ultrathin sections were collected and double contrasted (15 min with 13% uranyl acetate (Merck, Darmstadt, Germany) and 5 min with 2.8% lead citrate (Fluka, Buchs, Switzerland)) and examined with a JEOL JEM 100CX II transmission electron microscope (JEOL, Tokyo, Japan) operating at 80 kV, and relevant images were recorded with MegaView G3 camera (EMSIS, Münster, Germany).

### 2.6. Network Pharmacology Analysis

Targets of quercetin were identified using the following drug databases: the Drug–Gene Interaction database (DGIdb) [21], Drug Bank [22], PharmMapper [23], Swiss Target Prediction [24], UniprotKB [25], and BATMAN [26]. Genes involved in cardiotoxicity were obtained from GeneCards, DisGeNET [27], OMIM [28], and UniprotDB [25]. Common targets between quercetin and cardiotoxicity were identified using InteractiVenn [29]. Protein–protein interaction (PPI) networks were built in R version 4.4.0 (2024-04-24 ucrt)—“Puppy Cup” [30]—using the rbioapi [31] package with a high level of confidence. Pathway enrichment analysis of the common target genes between quercetin and cardiotoxicity was done using the enrichR [32] package with the hallmark gene sets from the Molecular Signatures Database (MSigDB) [33].

### 2.7. Statistical Analysis

Data were analyzed using the GraphPad Prism 8 (GraphPad Software, San Diego, CA, USA). Continuous variables were reported using mean values and standard deviation and graphically presented using boxplots and whiskers. Unless otherwise specified, whiskers were set to represent the minimum and maximum values of the measurements. Significance among groups was tested using a one-way ANOVA test. Multiple comparisons of the groups were done using Sidak’s multiple comparisons test. The significance threshold was set at a *p*-value of <0.050.

## 3. Results

### 3.1. Oral Quercetin Improves Heart Function and Echocardiographic Parameters

Initially, we examined the impact of chronic DOX treatment on the cardiovascular parameters of the rats. Serum NT-proBNP, CK-MB, and TnI levels [34] were markedly increased in DOX-treated rats, indicating the occurrence of cardiac damage (Figure 2A). In parallel, cardiac ultrasound revealed multiple changes in normal cardiac function and structure (Figure 2B). Chronic DOX treatments resulted in a significant reduction in the interventricular septal thickness in systole (DOX: 0.80 ± 0.14 vs. control: 1.43 ± 0.25; *p* < 0.001), LV posterior wall thickness in systole (DOX: 1.30 ± 0.30 vs. control: 2.23 ± 0.10; *p* < 0.001), LV ejection fraction (DOX: 74.20 ± 3.49 vs. control: 91.60 ± 3.21; *p* < 0.001) and fractional shortening (DOX: 37.7 ± 2.88 vs. control: 58.80 ± 5.99; *p* < 0.001). While not significant, a tendency towards a reduction in the interventricular septal thickness in diastole, LV end-diastolic dimension, LV end-systolic dimension, and LV posterior wall thickness in systole were also seen. These cumulative changes effectively confirmed the establishment of chronic doxorubicin-induced cardiotoxicity.

The administration of quercetin showed a noteworthy normalization in both serum markers of cardiac insult and ultrasound-based functional and structural parameters. Specifically, animals treated with quercetin at a 50 mg dose displayed a noticeable inclination towards the normalization of serum NT-proBNP levels. This tendency reached statistical significance upon doubling the quercetin dose (DOX/Q-100: 1170 ± 172 vs. DOX: 1644 ± 317; *p* = 0.045), effectively restoring normal levels of NT-proBNP (DOX/Q-100: 1170 ± 172 vs. control: 1205 ± 335; *p* > 0.990).

In terms of ultrasound parameters, significant improvements were observed in both LV ejection fraction and fractional shortening. Specifically, the administration of 100 mg of quercetin significantly enhanced cardiac function (LV ejection fraction: DOX/Q-100: 82.2 ± 4.02 vs. DOX: 74.2 ± 3.49; *p* = 0.009 and FS: DOX/Q-100: 47.60 ± 8.13 vs. DOX: 37.70 ± 2.88; *p* = 0.049). However, it is important to note that quercetin did not completely normalize the parameters to the levels seen in the control group (LV ejection fraction: DOX/Q-100: 82.2 ± 4.02 vs. control: 91.60 ± 3.21; *p* = 0.002 and FS: DOX/Q-100: 47.60 ± 8.13 vs. control: 58.80 ± 5.99; *p* = 0.020) (Figure 2B).

### 3.2. Network Pharmacology Analysis Yields Common Targets of Quercetin in Cardiotoxicity

We sought to explore the putative pathways involved in quercetin effects in DOX-induced cardiotoxicity by leveraging network pharmacology bioinformatic analysis. For this, we employed the use of multiple gene and drug databases and interaction predictors. The results showed a total of 112 gene targets that were in common across at least two of the databases. The PPI network revealed both physical and functional associations between proteins with an average node degree of 25.71. Enrichment analysis of these common targets revealed 35 hallmark pathways with an adjusted *p*-value < 0.05 (Figure 3B). Among these, the most significant were involved in apoptosis, reactive oxygen species (ROS) signaling, inflammatory and immune-related processes, and DNA damage (UV response) (Figure 3).

### 3.3. Quercetin Alleviates DOX-Induced Oxidative Stress

Considering these results, we further explored the pathological mechanisms underlying DOX-induced cardiac damage by assessing representative proteins and markers for ROS and antioxidant defense-related pathways, DNA damage, and apoptosis. We analyzed the levels of Nrf2, the master regulator of cellular redox homeostasis [35], and SOD1, an antioxidant enzyme that protects cells from oxidative stress [36]. Our findings revealed a notable elevation in Nrf2 levels (relative OD, DOX: 40.2 ± 8.29 vs. control: 25.30 ± 2.37; *p* = 0.040) and a simultaneous decline in SOD1 (DOX: 7.59 ± 1.53 vs. control: 18.60 ± 5.47; *p* = 0.048) following prolonged DOX administration. Furthermore, the administration of quercetin effectively reinstated Nrf2 levels to their baseline values, both in the case of the 50 mg dose (DOX/Q-50: 19.40 ± 8.77 vs. DOX: 40.20 ± 8.29; *p* = 0.009; DOX/Q-50: 19.40 ± 8.77 vs. control: 25.30 ± 2.37; *p* = 0.350) as well as the 100 mg dose DOX/Q-100: 21.60 ± 8.34 vs. DOX: 40.20 ± 8.29; *p* = 0.020; DOX/Q-100: 21.60 ± 8.34 vs. control: 25.30 ± 2.37; *p* = 0.560). Similar effects were seen in the case of SOD1 levels for the 50 mg dose (DOX/Q-50: 27.20 ± 9.63 vs. DOX: 7.59 ± 1.53; *p* = 0.003; DOX/Q-50: 27.20 ± 9.63 vs. control: 18.60 ± 5.47; *p* = 0.110) and the 100 mg dose (DOX/Q-100: 19.60 ± 3.41 vs. DOX: 7.59 ± 1.53; *p* = 0.040; DOX/Q-100: 19.60 ± 3.41 vs. control: 18.60 ± 5.47; *p* = 0.840) (Figure 4A).

Next, we explored potential DNA damage triggered by DOX-induced ROS by quantifying γH2AX and 8-hydroxy-2-deoxyguanosine (8-OHdG) [37,38]. The analysis suggested a trend toward elevated γH2AX (*p* = 0.220) and 8-OHdG (*p* = 0.480) levels in rats subjected to DOX, which appeared to be mitigated by quercetin administration (γH2AX: *p* = 0.220 for both concentrations; 8-OHdG: *p* = 0.900 and *p* = 0.460) (Figure 4B), effects which did not reach the threshold for significance. These changes were also accompanied by similar modifications in caspase-9 levels, both in the proactive and active (cleaved) forms (Figure 4C). Similarly, DOX-treated animals displayed a slight increase in both active and inactive forms (*p* = 0.220). This increase was only tendentially reduced by the addition of quercetin to the DOX treatment (*p* = 0.310 and *p* = 0.220, respectively).

### 3.4. Quercetin Restored Normal Myocardium Ultrastructure

TEM examination of myocardium samples from the control group revealed normal ultrastructure of cardiomyocytes. Myofibrils with a regular pattern of sarcomeres were separated by regions containing mitochondria and many small granules of glycogen, and by profiles of sarcoplasmic reticulum (Figure 5A). Mitochondria were elongated or round, with a homogeneous matrix and numerous cristae (Figure 5A,B). In the DOX group, the myofibrils were rarefied, fragmented, or even lysed on extensive regions (Figure 5C,D). A reduced amount of glycogen was noted, as well as proliferation and enlargement of the sarcoplasmic reticulum (Figure 5C,D). Among mitochondria with normal ultrastructure, swollen mitochondria devoid of cristae and with an electron-lucent matrix were also found (Figure 5D). Many secondary lysosomes were identified in the cardiomyocytes of this group (Figure 5D) (Table 1).

Cardiomyocytes from the DOX/Q50 group contained myofibrils with a regular pattern of the sarcomeres and myofilaments, but with some rarefied regions (Figure 6A). The sarcoplasmic reticulum with a normal aspect or with a lower diameter occupied large regions between myofibrils (Figure 6A,B). An important amount of small glycogen granules was present in these regions between myofibrils (Figure 6B), and in many cells among the myofilaments (insert of Figure 6A). A certain number of mitochondria was altered, showing a reduced number of cristae or no cristae at all in their matrix of normal density (Figure 6B). In cardiomyocytes from the DOX/Q100 group, the myofibrils retained a normal ultrastructure and pattern of sarcomeres (Figure 6C,D). The profiles of the sarcoplasmic reticulum appeared in lower numbers (comparable with those in the control group), either dilated (Figure 6C) or of normal diameter (Figure 6D)—which was prevalent in the examined sections.

A particular ultrastructural aspect identified in this group was represented by the presence of numerous large granules of glycogen (type 3) (Figure 6C). Elongated or round mitochondria showed mostly characteristic aspects (Figure 6C,D). A few of them were altered, containing a lower number of cristae (Figure 6D). Also, several secondary lysosomes were found in the cardiomyocytes of this group (Figure 6D) (Table 1). Regarding other representative ultrastructural features, endothelial cells of the endocardium and of capillaries from the myocardium did not show differences among the four considered groups. Also, no differences were observed in junctions forming intercalated discs.

## 4. Discussion

Anthracyclines, such as DOX, have revolutionized cancer treatment due to their potent antitumor properties. Despite substantial research efforts aimed at modifying anthracyclines to enhance efficacy and reduce undesired side effects, the persistent concern of cardiotoxicity has yet to be resolved [5,6]. The intricate and complex mechanisms underpinning anthracycline-induced cardiotoxicity, involve mechanisms such as oxidative stress, mitochondrial dysfunction, and DNA damage [39,40,41]. Despite many efforts to explore different treatment strategies, the goal of diminishing cardiotoxicity has yet to be fulfilled [42]. Thus, to help close this unmet clinical need, our study explored the effects of quercetin as a cardioprotective agent and whether it can help restore normal cardiac function in chronic DOX-treated rats. This objective stems from the well-established role of quercetin as a ROS scavenger, shown to have beneficial properties across various diseases, including other cardiovascular conditions [14,43,44], but insufficiently investigated in the context of DOX-induced cardiac damage.

### 4.1. Mechanisms of Quercetin Cardioprotective Effects in DOX Cardiotoxicity

Cardiomyocytes, due to their elevated mitochondrial volume constituting approximately 30% of the cell, are particularly vulnerable to oxidative damage [33]. The interaction of DOX with iron generates ROS. Additionally, DOX treatment has been shown to reduce the levels of endogenous antioxidant mechanisms, notably SOD1 [33]. SOD1 plays a vital role in the antioxidant defense system by converting superoxide radicals into hydrogen peroxide and molecular oxygen. The diminished SOD1 levels following anthracycline treatment contribute to the accumulation of superoxide radicals, intensifying oxidative damage to cellular components. In fact, SOD1 was significantly reduced in the heart lysates from chronic DOX-treated rats. Moreover, our previous results showed that this decrease was present from the first dose of DOX [45]. This was accompanied by a significant adaptative increase in Nrf2 levels (Figure 7), a well-known player in DOX-mediated cardiotoxicity [46]. These pronounced imbalances were efficiently restored by quercetin administration, most likely due to its ROS scavenging capabilities. While ROS drugs are known to trigger Nrf2 signaling, an adaptative pathway for antioxidant enzyme upregulation, Nordgren et al. showed that chronic DOX administration in rats resulted in a loss of Nrf2 causing a persistent increase in oxidative stress and mitochondrial damage [47,48]. This apparently opposite result could be attributed to the fact that in their experimental design, rats were allowed a “five-week drug-free holiday” after the 2-week DOX treatment, thus allowing the installment of this Nrf2 rebound [47]. One might hypothesize that this imbalance could be mitigated through the administration of quercetin, which can prevent the depletion of antioxidant enzymes and maintain the homeostasis of redox systems.

Albeit statistically insignificant, it is worth discussing that DOX-treated hearts showed a tendency towards double-strand DNA breaks, suggested by the increase in γH2AX. This was accompanied by an increase in 8-OHdG (a well-known biomarker indicating oxidative damage to DNA) which is produced following the restoration of DNA damage caused by ROS [37]. This mechanism is important for the antitumoral effect of DOX in cancer cells [49]. Quercetin administration showed a tendency to reduce both γH2AX and 8-OHdG (Figure 7). Thus, one might theorize that this phenomenon could possibly hinder the antitumoral effects of DOX. Encouragingly, some studies showed that quercetin also had a synergistic effect with DOX-based chemotherapy in cancer models [50,51,52].

The benefits of quercetin use are multiple, some of which are supported by studies with high confidence levels. For example, a systematic review of in vivo studies showed that quercetin reduced the fat mass index in rats, and prevented dyslipidemia, showing a potential antiatherosclerotic effect [53,54]. Moreover, evidence showed a reduction in blood pressure in vivo. This positive effect has also been observed in humans, as demonstrated in a meta-analysis of randomized controlled trials [55]. The systematic review highlighted that a shorter intervention time and a higher dosage were more effective in restoring cardiac function [54]. While we did not explore the potential effects of different lengths of quercetin treatment, we noticed that the 100 mg/kg/day quercetin dose showed more significant improvement than the 50 mg/kg/day dose. This was particularly evident in the case of LVEF and FS, where the 50 mg/kg/day dose resulted in a 2.16% improvement in cardiac output with respect to DOX, while the 100 mg/kg/day dose resulted in a more substantial improvement of 10.78%, the latter surpassing the threshold for statistical significance. Moreover, the higher dose also showed a significant reduction in serum NT proBNP, suggesting a reduction in cardiac distress. Although the lower dose also showed significant alleviation at molecular and ultrastructural levels, only the 100 mg dose resulted in a functional improvement which is the most important outcome for patients. To the best of our knowledge, this is the first study to show that quercetin increased the cardiac output in chronic DOX treatment in vivo.

While numerous studies explored the pathways responsible for DOX-induced cardiac damage and the associated histological changes, less research seems to have specifically addressed the functional aspects of the heart. These functional aspects are of paramount importance, as the ultimate measure of significance lies in the heart’s ability to function normally. Thus, we explored the potential of quercetin to improve the well-documented alterations in US parameters induced by DOX. We showed that quercetin substantially reinstated various measurements, with some returning entirely to the levels observed in the control group, while others remained slightly below the normal values. A similar study, although not on DOX-induced cardiac damage, showed that administration of intravenous quercetin was associated with less reperfusion-induced intramyocardial hemorrhage in patients with myocardial infarction. However, no significant differences in LEFV and LV remodeling indicators were noted [56]. It is noteworthy to highlight that the context is different, given that myocardial infarction is an acute event compared to the chronic nature of the remodeling induced by DOX. As a result, the mechanisms involved may differ and act at distinct timings. However, these favorable outcomes are particularly encouraging, considering that the cardiac damage induced by DOX is irreversible. Therefore, patients are in need of a drug to enhance their cardiac output. Even a marginal improvement facilitated by quercetin could have a substantial impact on both the lifespan and quality of life of these patients. Other studies investigated the cardioprotective effects of quercetin in DOX-induced cardiotoxicity and also found a significant decrease in serum markers for cardiac injury in rats receiving a dose of 80 mg/kg quercetin [3]. Some works have shown that quercetin significantly facilitated cell survival by maintaining cell morphology and rearranging the cytoskeleton, thus preventing ischemia and reperfusion injury in cardiomyocytes. The authors postulated that pretreating with quercetin might scavenge ROS and diminish oxidative stress [57,58]. Others showed that prophylactic quercetin administration increased cell viability and antioxidant proteins and restored myocardial energy metabolism, thus preventing the deleterious effects of DOX [59,60].

### 4.2. Ultrastructural Changes Associated with Quercetin in DOX Cardiotoxicity

While studies mostly focused on histopathological changes in heart tissues [3,50,61], the TEM examination of endomyocardial biopsy holds significant additional diagnostic value, particularly in patients suspected of non-ischemic cardiomyopathy [62]. Histopathologic analysis showed a marked reduction in the percentage of degenerative myocardial cells, a decrease in inflammatory cells, and an area of fatty infiltration and blood vessel congestion in the quercetin-treated rats [3]. These findings come to complement our TEM observations: DOX-treated animals had rarefied and severely damaged cardiomyocytes, with loss of myofibrils and distention of sarcoplasmic reticulum, changes which are well-known in DOX-induced cardiomyopathy [61]. In contrast, the ultrastructure of the cardiomyocytes was similar to that of the control group, especially in the case of the higher 100 mg dose of quercetin. These important ultrastructural changes could explain the loss of cardiac muscle contractility as demonstrated by the significant decrease in US parameters.

### 4.3. Study Limitations, Strengths, and Future Directions

It is important to note certain limitations that could impact the interpretation of our findings. Firstly, we did not directly measure ROS and relied solely on observing their indirect effects. For example, we evaluated the change in Nrf2 protein abundance as an indicator of oxidative stress and activation of cellular defense mechanisms without focusing on its activating post-translational modifications. Further detailed research could explore the mechanisms of Nrf2 activation in this context and evaluate the alternative activation pathways. Additionally, the borderline significance of some variables, such as the molecular study of DNA damage and apoptosis, suggests that our sample size of rats might have been insufficient to achieve the necessary statistical power for detecting significance, especially when applying the correction for multiple comparisons. Next, our study focused on the effects of quercetin on chronic DOX-induced cardiotoxicity without evaluating the effects of quercetin alone, as its benefits have been previously explored [63]. Nevertheless, our research provides valuable insights and paves the way for more comprehensive studies in the future aimed at further dissecting the effect of quercetin at a molecular level. The study’s strengths lie in the consistent and homogeneous treatment of animals, overseen by experienced cardiologists, ensuring uniformity in experimental conditions and robust data analysis. Furthermore, the use of multiple replicates in ultrasound parameter measurements adds a layer of reliability and repeatability to our results, enhancing the overall credibility of the study.

## 5. Conclusions

Our study provides novel evidence that the oral administration of quercetin effectively mitigated the cardiotoxic effects of chronic DOX treatment, as shown by the decrease in serum markers of cardiac injury. Additionally, we identified a previously unreported positive influence on cardiac output, highlighting the potential of quercetin to improve cardiac function. These findings strongly suggest that quercetin holds potential cardioprotective properties, revealing quercetin as a potential therapeutic intervention in clinical settings against DOX-induced cardiotoxicity.

## Figures and Tables

**Figure 1 antioxidants-13-01068-f001:**
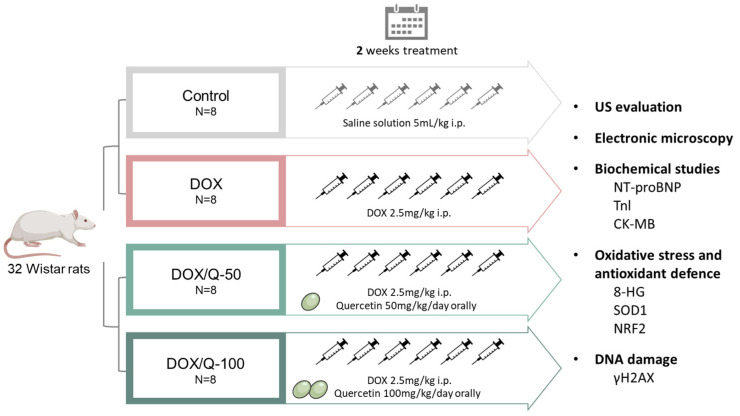
Study design and workflow. A total of 32 Wistar rats were divided into 4 groups: control, receiving i.p. injections of 0.9% saline solution; DOX, receiving i.p. injections with 2.5 mg/kg body-weight doxorubicin; DOX/Q-50, receiving i.p. injections with 2.5 mg/kg body-weight doxorubicin and 50 mk/kg/day quercetin orally; DOX/Q-100, receiving i.p. injections with 2.5 mg/kg body-weight doxorubicin and 100 mk/kg/day quercetin orally. DOX: doxorubicin; i.p.: intraperitoneal; Q: quercetin; NT-proBNP: natriuretic peptide; TnI: troponin I; LDH: lactate dehydrogenase; CK: creatine kinase; CK-MB: creatine kinase isoenzyme MB; SOD: superoxide dismutase.

**Figure 2 antioxidants-13-01068-f002:**
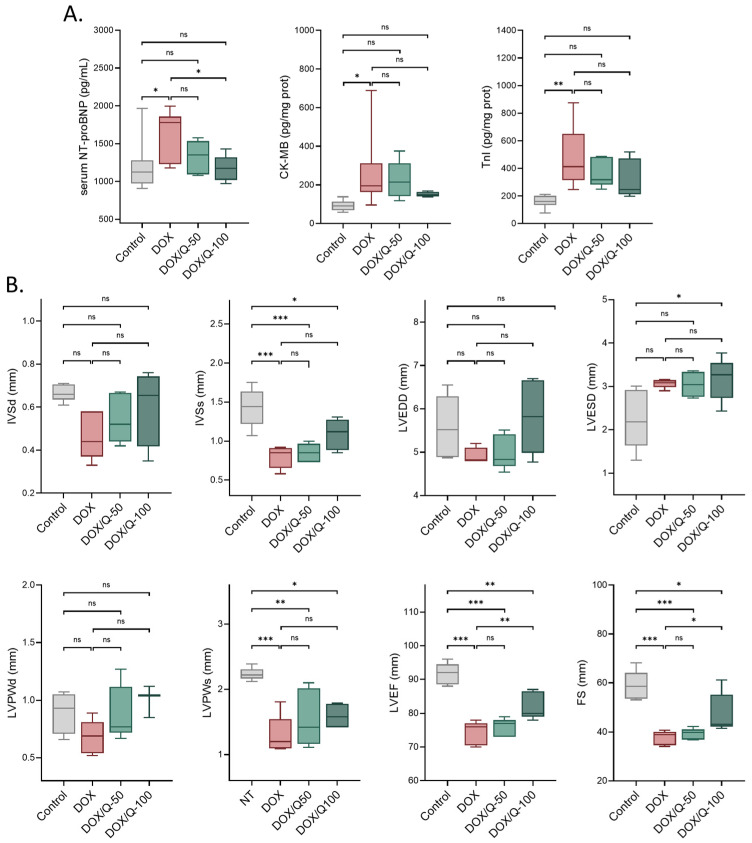
Oral administration of quercetin alleviates serum markers of cardiac injury and US parameters in an in vivo model of DOX-induced chronic cardiotoxicity. (**A**) NT-proBNP, CK-MB, TnI were significantly increased by multiple DOX doses in Wistar rats. Quercetin administration lowered the levels of these markers in a dose-dependent manner. (**B**) DOX altered the cardiac echocardiography parameters while quercetin improved cardiac function. DOX: doxorubicin; Q: quercetin; IVSd: interventricular septal thickness in diastole; IVSs: interventricular septal thickness in systole; LVPWs: LV posterior wall thickness in diastole; LVPWd: LV posterior wall thickness in systole; LVEDD: LV end-diastolic dimension; LVESD: LV end-systolic dimension; LVEF: LV ejection fraction; FS: fractional shortening. ** p* < 0.05; ** *p* < 0.01; *** *p* < 0.001.

**Figure 3 antioxidants-13-01068-f003:**
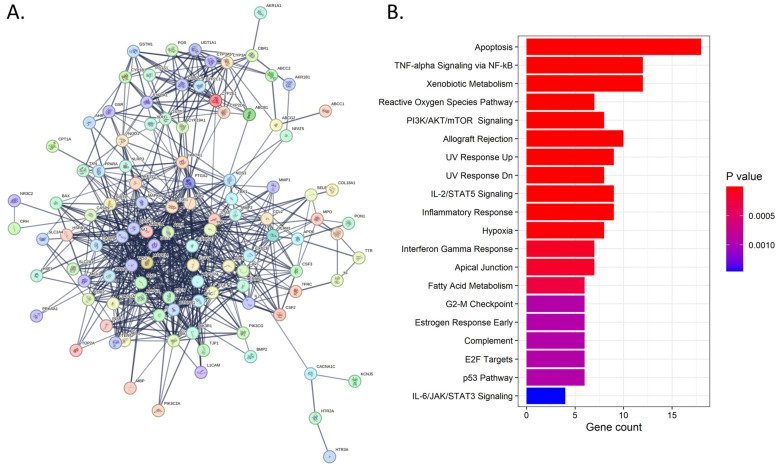
Network pharmacology analysis. (**A**). Protein–protein interaction map of the putative proteins commonly involved in quercetin effects and cardiotoxicity. (**B**). Barplot showing top 20 significant pathways from the functional enrichment.

**Figure 4 antioxidants-13-01068-f004:**
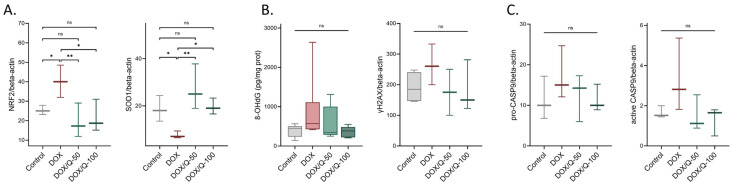
Oxidative damage in DOX and quercetin-treated rats. Alterations in the antioxidant defense system (**A**), DNA damage (**B**), and apoptosis (**C**) associated with DOX and quercetin treatments. Chronic DOX treatment induced an increase in Nrf-2 levels together with a depletion of SOD-1 enzyme indicating an alteration in ROS homeostasis. Quercetin treatment successfully reversed these changes. DOX: doxorubicin; Q: quercetin; Nrf2: Nuclear factor erythroid 2-related factor 2; SOD1: superoxide dismutase 1; γH2AX: H2A histone family member X; 8-OHdG: 8-Hydroxy-2-Deoxyguanosine. * *p* < 0.05; ** *p* < 0.01.

**Figure 5 antioxidants-13-01068-f005:**
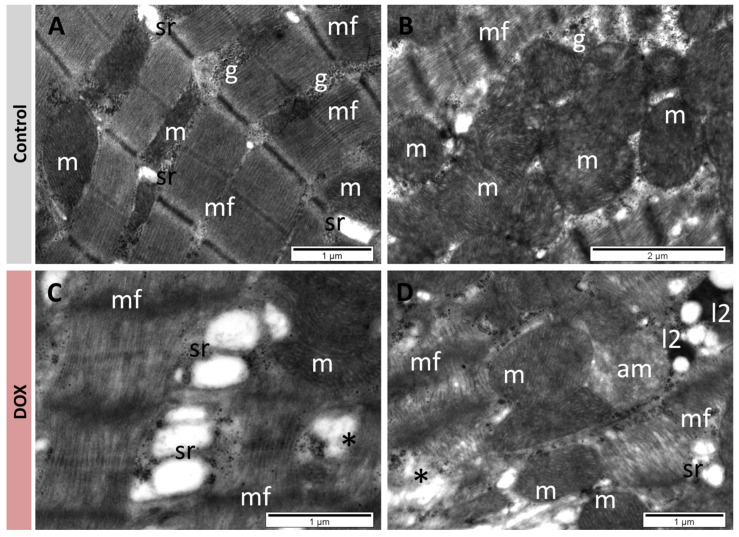
TEM images showing cardiomyocytes with normal ultrastructure in control group (**A**,**B**), and with modified ultrastructure in DOX group (**C**,**D**). Rarefaction and lesions of myofibrils; enlargement of sarcoplasmic reticulum, alteration of mitochondria, and presence of secondary lysosomes. am: altered mitochondria; g: glycogen; l2: secondary lysosomes; m: mitochondria; mf: myofibrils, sr: sarcoplasmic reticulum; *: lysed myofibrils.

**Figure 6 antioxidants-13-01068-f006:**
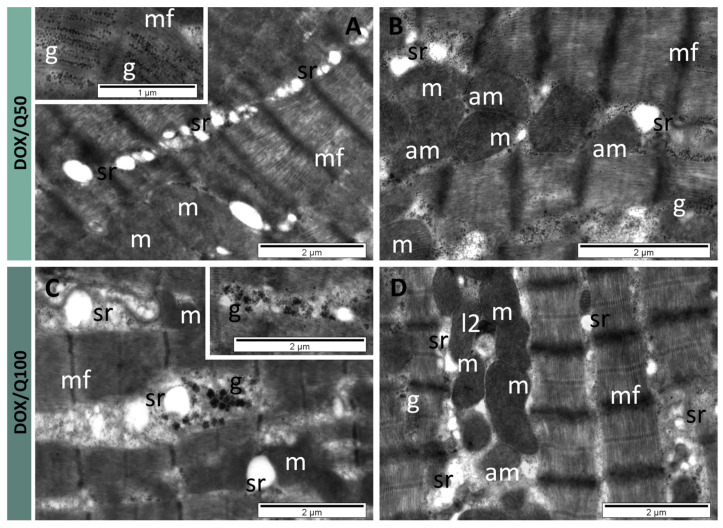
TEM images showing cardiomyocytes with slightly modified ultrastructure in DOX/Q50 group (**A**,**B**). Proliferated sarcoplasmic reticulum, rarefied myofibrils, numerous granules of glycogen and altered mitochondria, and cardiomyocytes with nearly normal ultrastructure in DOX/Q100 group (**C**,**D**), Normal aspect of myofibrils; rare, dilated profiles of sarcoplasmic reticulum; presence of large granules of glycogen; rare, altered mitochondria; and rare secondary lysosomes. am: altered mitochondria; g: glycogen; l2: secondary lysosomes; m: mitochondria; mf: myofibrils, sr: sarcoplasmic reticulum.

**Figure 7 antioxidants-13-01068-f007:**
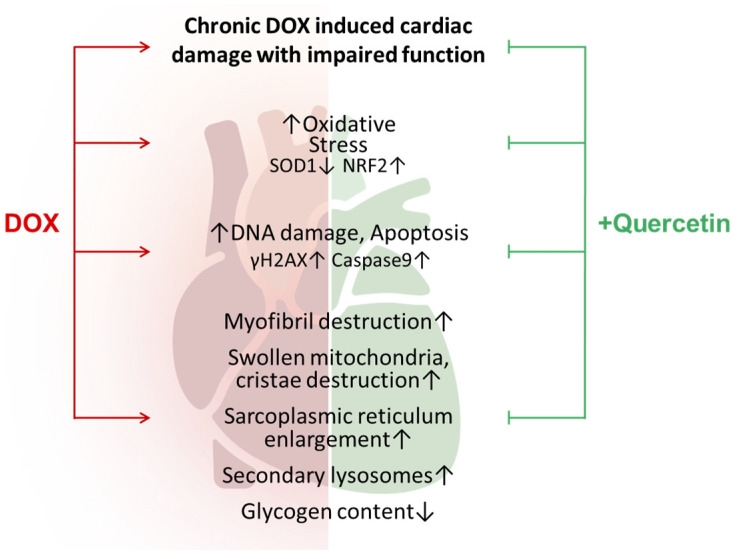
Proposed mechanisms of action of quercetin treatment in chronic DOX-induced cardiotoxicity. Red arrows indicate the damaging effects of DOX which are alleviated by quercetin administration (green blunt arrows).

**Table 1 antioxidants-13-01068-t001:** Summary of the cardiomyocyte alterations found at TEM in the four groups.

	Control	DOX	DOX/Q-50	DOX/Q-100
**Cardiomyocyte ultrastructure**	normal	altered	slightly modified	nearly normal
**Myofibrils**	normal	rarefied, fragmented, and lysed	sarcomeres with regular pattern; some rarefied regions	normal aspect and pattern of sarcomeres
**Mitochondria**	elongated or round; homogeneous matrix; numerous cristae	swollen mitochondria; devoid of cristae	altered; reduced number of cristae or no cristae; matrix of normal density	elongated or round; rare altered, with lower number of cristae
**Sarcoplasmic reticulum**	normal	enlarged	normal aspect + with lower diameter	mostly normal; rare dilated
**Glycogen**	many small granules	reduced	many small granules	numerous large granules of glycogen
**Secondary lysosomes**		many		several

## Data Availability

Raw data can be made available upon reasonable request from the corresponding authors.

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
