# Peer review of "Mitigating Doxorubicin-Induced Cardiotoxicity through Quercetin Intervention: An Experimental Study in Rats"

_antioxidants, 2024, doi:10.3390/antiox13091068_

Round 1
Reviewer 1 Report
The manuscript offers information on the potential cardioprotective effects of quercetin 50 and 100 mg (Q), after chronic treatment with DOX, in 32 rats, for two weeks. Cardiac ultrasound (US), markers of cardiac and oxidative damage—NT pro-BNP, troponin I and CK-MB, SOD1, NRF2, DNA damage—helped prove the future oral administration of Q to ameliorate DOX-induced cardiotoxicity.
The presented study is innovative, original and relevant to the field of DOX-induced cardiotoxicity. The manuscript is written in standard English, with minor stylistic and grammatical changes required.
Small remarks.
1. The proposed abstract devotes more attention to DOX than to quartzetin priorities. To rewrite.
2. Submission of a graphic abstract is mandatory.
3. paragraph 34-4o to be shortened
4. Anthracycline-induced cardiotoxicity is categorized into two distinct types: (i) Type I, and (ii) Type II ...to be presented schematically.
5. Quercetin (QRN) ... and potential protective effects of Q.... to standardize the abbreviation in the text.
6. Study design...line 88-99 to shorten and present concisely.
7. Biochemical and western blot assays - all used markers should be given on a new line; to unify font; to shorten
8. TEM analysis - to be shortened
9. when describing results, a lot of abbreviations accumulate and this complicates the readability - where it is possible to avoid
10. 3.4. Quercetin restored normal myocardium ultrastructure. - the disabilities should be presented in a table for better readability
11. Which of the two concentrations of quartzetin is better? Probably why? To present a probable mechanism of controlled cardiotoxicity/
12. Limitations and future prospects of the research are missing.
13. The conclusion is general - to specify the innovativeness of the research.
14. 50% of used articles with aot of the last 5 years; references 20, 35 to be replaced.
The manuscript offers information on the potential cardioprotective effects of quercetin 50 and 100 mg (Q), after chronic treatment with DOX, in 32 rats, for two weeks. Cardiac ultrasound (US), markers of cardiac and oxidative damage—NT pro-BNP, troponin I and CK-MB, SOD1, NRF2, DNA damage—helped prove the future oral administration of Q to ameliorate DOX-induced cardiotoxicity.
The presented study is innovative, original and relevant to the field of DOX-induced cardiotoxicity. The manuscript is written in standard English, with minor stylistic and grammatical changes required.
Small remarks.
1. The proposed abstract devotes more attention to DOX than to quartzetin priorities. To rewrite.
2. Submission of a graphic abstract is mandatory.
3. paragraph 34-4o to be shortened
4. Anthracycline-induced cardiotoxicity is categorized into two distinct types: (i) Type I, and (ii) Type II ...to be presented schematically.
5. Quercetin (QRN) ... and potential protective effects of Q.... to standardize the abbreviation in the text.
6. Study design...line 88-99 to shorten and present concisely.
7. Biochemical and western blot assays - all used markers should be given on a new line; to unify font; to shorten
8. TEM analysis - to be shortened
9. when describing results, a lot of abbreviations accumulate and this complicates the readability - where it is possible to avoid
10. 3.4. Quercetin restored normal myocardium ultrastructure. - the disabilities should be presented in a table for better readability
11. Which of the two concentrations of quartzetin is better? Probably why? To present a probable mechanism of controlled cardiotoxicity/
12. Limitations and future prospects of the research are missing.
13. The conclusion is general - to specify the innovativeness of the research.
14. 50% of used articles with aot of the last 5 years; references 20, 35 to be replaced.
Author Response
Dear peer reviewer,
please find our point by point responses in the attached document. Thank you for your valuable input! We highly appreciated the useful feedback!
Yours,
Camelia

Reviewer 2 Report
The manuscript by Dulf et al. investigates the cardioprotective effects of two different doses of quercetin (Q) on doxorubicin (DOX)-induced cardiotoxicity in vivo utilizing rats. The study involves evaluations of cardiac function and oxidative stress, examination of myocardial tissue by transmission electron microscopy, and biochemical and network pharmacology analyses. The paper explores an important topic, i.e., countering detrimental effects of DOX treatment on the myocardium, however there are some concerns.
Major:
1) Considering Q have been investigated in the context of DOX-induced cardiomyopathy, the authors should highlight the new information provided in the study more clearly. The paper would be stronger if the novelty is stated better.
2) “Q only” groups should be included in the study for comparison.
3) Although Nrf2 and SOD1 are antioxidant defenses, their protein levels cannot be utilized to determine changes in oxidative stress (or used as “ROS markers” as stated in Figure 4 caption). ROS levels need to be determined by employing one of the widely accepted methods for ROS detection.
4) The paper will be stronger if Nrf2 activation is examined in addition to its protein levels.
5) As the authors acknowledge, the data regarding DNA damage and apoptosis do not have statistical significance (Figure 4). Since they constitute an important part of the manuscript, stronger evidence needs to be provided.
6) In section 3.3, the description of results related to caspase needs to be elaborated.
Minor:
1) In Figure 1, SOD1 and Nrf2 are placed under oxidative stress, which is not an accurate description. They are antioxidant defenses.
2) Section 3.1 and Figure 2: Based on the authors’ description of significance in the statistical analysis section (p<0.05), p=0.05 does not indicate statistical significance. Please clarify.
Author Response
Dear peer reviewer,
please find our point by point responses in the attached document. Thank you for your valuable time and input! We highly appreciated the constructive feedback! We feel that the manuscript has been improved thanks to your suggestions.
Yours,
Camelia

Round 2
Reviewer 1 Report
-
-
Author Response
Dear Reviewer,
We would like to thank you again for your previous comments. I am afraid that I cannot find the new feedback you have given for the second round revision. I will give a notice to the Editors as well should some type of error have occured.
Yours,
Camelia
Reviewer 2 Report
In the revised manuscript, the authors have partially addressed my concerns, which have strengthened the paper. However, there are a couple of remaining issues which require further attention.
1) “Materials and Methods” section: Unfortunately, using the term "activation of Nrf2" results in an inaccurate statement. Nrf2 expression levels were determined, showing the upregulation of Nrf2. Either data showing Nrf2 activation should be included, or the lack of these experiments should be stated in the “study limitations” section.
2) Although the authors have addressed the critique regarding the lack of “quercetin (Q) only” groups in the cover letter, the issue and its justification should be included in the “study limitations” section also.
In section 3.3, “the influence of oxidative stress alterations” is a very vague term and it confuses the reader. The authors analyze antioxidant defenses (and not ROS levels or oxidative stress) and it should be stated as such. The potential interpretation of the observations can be mentioned (including the corresponding references).
Author Response
Dear peer-reviewer,
Thank you for going through our paper and for the prompt feedback. Please find attached our responses to the remaining issues. We hope to have addressed them fully. We look forward to your response!
